# Prospective Observational Study to Evaluate the Effect of Different Levels of Positive End-Expiratory Pressure on Lung Mechanics in Patients with and without Acute Respiratory Distress Syndrome

**DOI:** 10.3390/jcm9082446

**Published:** 2020-07-31

**Authors:** Mascha O. Fiedler, Dovile Diktanaite, Emilis Simeliunas, Maximilian Pilz, Armin Kalenka

**Affiliations:** 1Department of Anesthesiology, Heidelberg University Hospital, 69120 Heidelberg, Germany; mascha.fiedler@med.uni-heidelberg.de (M.O.F.); diktanaite@gmail.com (D.D.); simeliunui@gmail.com (E.S.); 2Department of Anesthesiology and Intensive Care Medicine, District Hospital Bergstrasse, 64646 Heppenheim, Germany; 3Department of Anesthesiology, Lucerne Hospital, 6000 Lucere, Switzerland; 4Institute of Medical Biometry and Informatics, University of Heidelberg, 69120 Heidelberg, Germany; pilz@imbi.uni-heidelberg.de; 5Faculty of Medicine, University of Heidelberg, 69120 Heidelberg, Germany

**Keywords:** acute respiratory distress syndrome, lung physiology, mechanical ventilation, end-expiratory lung volume, esophageal pressure, transpulmonary pressure, lung injury, positive end-expiratory pressure

## Abstract

Background: The optimal level of positive end-expiratory pressure is still under debate. There are scare data examining the association of PEEP with transpulmonary pressure (TPP), end-expiratory lung volume (EELV) and intraabdominal pressure in ventilated patients with and without ARDS. Methods: We analyzed lung mechanics in 3 patient groups: group A, patients with ARDS; group B, obese patients (body mass index (BMI) > 30 kg/m^2^) and group C, a control group. Three levels of PEEP (5, 10, 15 cm H_2_O) were used to investigate the consequences for lung mechanics. Results: Fifty patients were included, 22 in group A, 18 in group B (BMI 38 ± 2 kg/m^2^) and 10 in group C. At baseline, oxygenation showed no differences between the groups. Driving pressure (ΔP) and transpulmonary pressure (ΔP_L_) was higher in group B than in groups A and C at a PEEP of 5 cm H_2_O (ΔP A: 15 ± 1, B: 18 ± 1, C: 14 ± 1 cm H_2_O; ΔP_L_ A: 10 ± 1, B: 13 ± 1, C: 9 ± 0 cm H_2_O). Peak inspiratory pressure (P_insp_) rose in all groups as PEEP increased, but the resulting driving pressure and transpulmonary pressure were reduced, whereas EELV increased. Conclusion: Measuring EELV or TPP allows a personalized approach to lung-protective ventilation.

## 1. Background

Invasive ventilation, one of the most frequently applied strategies in the intensive care unit is often lifesaving. Nonetheless, it is a potentially harmful intervention that may lead to ventilator-induced lung injury (VILI) [1]. Causes of VILI are mainly stress, transpulmonary pressure (TPP) and strain and applied tidal volume (V_T_) in relation to end-expiratory lung volume (EELV) [2]. These two elements are not routinely measured at the bedside in patients with acute respiratory distress (ARDS) [3]. Therefore, most guidelines emphasize using lower V_T_ (4–8 mL/kg predicted body weight) and lower inspiratory pressures (plateau pressure <  28 cm H_2_O) [4,5] as well as a positive end-expiratory pressure (PEEP) based on oxygenation [6], as a standard of care.

Although low tidal volume ventilation in patients with ARDS improves survival, when compared with high tidal volume [7], this difference between a low and intermediate tidal volume could not be detected in patients without ARDS [8]. Airway pressure alone may also be insufficient to guide lung-protective PEEP titration in ARDS. The contribution of chest wall and abdomen to pleural pressure and respiratory system mechanics is unpredictably in the critically ill patients [9,10]. Approximately 50% of the intra-abdominal pressure (IAP) is transmitted to the intrathoracic compartment [11]. Therefore, it has a direct impact on EELV and TPP. 

Despite this, the relationship between transpulmonary pressure, lung volume and IAP in different types of patients with mechanical ventilation has not yet been fully explored.

We analyzed the relationship of EELV, lung mechanics and IAP in 3 patient groups: group A—patients with ARDS, group B—patients with obesity and group C—a control group. Three levels of PEEP (5, 10, 15 cm H_2_O) were used to investigate the consequences of the different PEEP levels for lung mechanics. We hypothesized that an advanced measurement of EELV, TPP and IAP would allow an individualized titration of mechanical ventilation. 

## 2. Methods

### 2.1. Trial Design

This project was performed in accordance with the Declaration of Helsinki and the approval of the local Ethics Committee of the Heidelberg Medical Faculty at the University of Heidelberg (S−579/2016; 15.02.2017), between April 2017 and January 2019, in the intensive care unit of Hospital Bergstrasse. The study protocol was registered in the German Clinical Trials Register (DRKS00012639). After providing appropriate information, written consent to study participation was obtained from the participating patients or their family members, which could be revoked without giving reasons. 

All consecutive mechanically ventilated adult patients admitted who expected not to be extubated within 48 h were examined for possible inclusion in the study. The planed study protocol included measurements over a maximum of 3 days. Exclusion criteria were start of ventilation over 24 h previously, severe hemodynamic instability, higher grade of chronic obstructive lung disease (COPD) with GOLD grade 3 and 4, acute exacerbation of COPD, lung emphysema, pulmonary arterial hypertension, suspected elevated pulmonary arterial pressure in the echocardiography, application of a PEEP higher than 20 cm H_2_O, an inspired oxygen concentration (F_i_O_2_) ≥ 0.8, severe ARDS with a ratio between partial arterial pressure of oxygen to F_i_O_2_ (P/F ratio) < 100), an extracorporeal gas-exchange procedure (ECMO), use of high-frequency oscillation ventilation (HFOV) and abnormal airway anatomy due to partial lung resection or fistulas. 

We predefined a priori cohorts as: Group A: patients with ARDS [12], group B: obese patients with the body mass index (BMI) > 30 kg/m^2^ without ARDS, group C: a control group comprising patients suffering from cardiac arrest or with a neurological disease without ARDS or BMI > 30 kg/m^2^.

All patients were ventilated in pressure-controlled mode with volume guaranty (Carescape R860, GE Healthcare, Madison, WI, USA) on the first day of the examination. Inspiratory pressure was controlled so that a tidal volume of 6 mL/kg predicted body weight (pbw) was reached. It was verified that the respective patient had been in supine position for at least 15 min before the start of each measurement, in order to ensure identical examination conditions. Patients were sedated and paralyzed with 0.6 mg/kg pbw rocuronium bromide for the time indicated in the study protocol.

### 2.2. Measurements and Calculations

A polyethylene catheter (Nutrivent multifunction nasogastric catheter, Sidam, San Giacomo Roncole, Italy) was used to measure esophageal pressures. Appropriate catheter position was confirmed as previously described [13] with a non-stress minimal volume implementation [14]. Tidal volume (V_T_), Peak inspiratory airway pressure (P_Insp_), PEEP, inspiratory esophageal pressure (P_EsInsp_) and end-expiratory esophageal pressure (P_EsExp_) were recorded from the ventilator. Values for ΔP and transpulmonary pressure (ΔP_L_) were calculated as previously described [15]. Transpulmonary inspiratory pressure (TPP_Insp_) was calculated as TTP_Insp_ = P_Insp_ − P_EsInsp_ and transpulmonary expiratory pressure (TPP_Exp_) as TPP_Exp_ = PEEP − P_EsExp_. Elastance of the respiratory system (E_RS_) was calculated as E_RS_ = (P_Insp_ − PEEP)/V_T_, chest wall elastance (E_CW_) as E_CW_ = (P_EsInsp_ − P_EsExp_)/V_T_ and elastance of the lung (E_L_) as E_L_ = E_RS_ − E_CW_. We measured EELV at the bedside as previously described without interrupting mechanical ventilation at the designated PEEP level [16]. We measured EELV as an absolute value as calculated by the ventilator. This absolute EELV was then normalized by dividing by the predicted body weight of the patient. C_Stat_ was measured by the ventilator during an inspiratory hold. End-expiratory IAP (IAP_Endex_) was measured as recommended [17] and zeroed at the midaxillary level.

### 2.3. Study Protocol

The measurement of EELV, airway, esophageal and abdominal pressures at a PEEP of 15 cm H_2_O: The patient was ventilated for at least 15 min at a PEEP of 15 cm H_2_O. Achieving a steady-state situation by means of a stable carbon dioxide volume (VCO_2_) for at least 10 min is essential for EELV measurement, as VCO_2_ is one of the main parameters for EELV calculation [16]. Therefore, patients with an extracorporeal CO_2_ elimination device were not included. EELV measurement was started on the R860 Carescape ventilator. The ventilator was equipped with a COVX module providing the data for EELV calculation. For EELV measurement, we used a stepwise change in F_i_O_2_ of 0.2. EELV at each PEEP and measured this twice (wash-out and wash-in). At a F_i_O_2_ increase of 0.2, each complete wash-out and wash-in cycle took about 40 breaths; therefore, one measurement was completed within 10 min. Values for P_Insp_, PEEP, P_EsInsp_ and P_EsExp_ were recorded from the ventilator. The IAP_Endex_ was measured in mmHg and converted into cm H_2_O.

After completion of the measurement program at a PEEP of 15 cm H_2_O, PEEP was reduced to 10 cm H_2_O, and inspiratory pressure was maintained to keep the tidal volume at 6 mL/kg pbw. Measurements at a PEEP of 10 cm H_2_O in analogy to PEEP 15 cm H_2_O were followed by PEEP reduction to 5 cm H_2_O.

#### 2.3.1. Measurement Errors and Cancellations

Respiratory reasons: drop of arterial oxygen saturation > 10%, decrease in tidal volume > 20%; cardio-circulatory reasons: drop of mean arterial pressure > 20%, change in heart rate > 20%; measurement error: if five consecutive measurements were aborted by the device, this was documented as a measurement error.

#### 2.3.2. Premature Study Exclusions

Reasons for exclusion of a patient prior to planed study end at day 3: sufficient spontaneous breathing, extubation, death. Persistent hemodynamic instability preventing measurements for several consecutive days.

### 2.4. Statistical Analysis

The collected data were analyzed using SPSS software (IBM SPSS Statistics 25, Release 08.2015 IBM, Armonk, NY, USA) and the statistical software R, version 3.5 (R Foundation for Statistical Computing, Vienna, Austria). Mean values and standard deviations were calculated for quantitative variables. We analyzed data in the three groups with ANOVA, and in case of statistical differences, we used the independent *t* test to capture the differences between three groups. We compared all groups in pairs: ARDS with obesity and then each group with the control group (A vs. B, B vs. C and A vs. C). We used the paired *t* test to assess the change of lung parameters in the same group at the different levels of PEEP (PEEP 15 vs. 5 cmH_2_O). We used a linear mixed model to evaluate the influence of lung mechanics, intraabdominal pressure and body mass index on each of these. A linear mixed model is a more complex method for analyzing the correlation between the parameters than the “classical” Pearson correlation and can process diverse variables and include random and fixed effects. Since the patients were measured repeatedly, we chose the patient as a random effect and PEEP as well as the measurement number as fixed effects. For each comparison, the two variables to compare were included as outcome variable and fixed effect, respectively. The result of a mixed model is an effect value with a corresponding *p*-value. The larger the effect size and, thus, the smaller the *p*-value, the stronger is the relationship between the two variables of interest. For all statistical tests, the significance level was set at *p* < 0.05. We used GraphPad PRISM 7 Software results (GraphPad Software, Inc., San Diego, CA, USA) for rerunning data and developing graphs, without differences in the statistical. The results are shown as mean ± standard error of the mean (SEM).

## 3. Results

Between April 2017 and January 2019, we included 50 patients (mean age 68 ± 13; 31 males, 19 females) in the study. Twenty-two were mechanically ventilated because of ARDS and were enrolled in group A. Group B consisted of 18 obese patients but without ARDS. In group C (control) we enrolled 10 patients, who had neither ARDS nor were obese (Figure 1). Main baseline characteristics of patients at day 1 are presented in Table 1. Weight and BMI were highest in Group B, with no differences between group A and C. Patients were ventilated with a V_T_ between 6 and 7 mL/kg pbw. P/F ratio was not significantly different in the three groups.

Respiratory mechanics measured in the study are presented in Table 2. Overall, we measured 399 values over time and at different PEEP levels (5, 10 and 15 cm H_2_O). This means that we had a drop out of 51 values (26 because of extubation, sufficient spontaneous breathing or death, 25 because of drop of arterial oxygen saturation >10% or a decrease in tidal volume > 20% (Appendix A). Peak inspiratory pressure, ΔP and ΔP_L_ were higher in obese patients (group B) compared to patients with ARDS (group A). Although P_Insp_ and ΔP did not differ between patients with ARDS patients and group C the resulting ΔP_L_ were significantly higher in group A than in group C. Obese patients showed higher P_Insp_, ΔP and ΔP_L_ than group C. The EELV were lower in obese patients than in group A and C. Although PEEP 15 cm H_2_O in comparison to PEEP 5 cm H_2_O increases P_Insp_ in all groups, the resulting ΔP and ΔP_L_ were significantly lower in all groups. The E_RS_ and E_L_ decreased with increasing PEEP in all groups, whereas E_CW_ was unchanged. Obese patients had higher E_RS_, E_L_ and IAP than patients in Group A and C. Increasing PEEP had no influence on IAP but increased P_EsExp_ in all groups.

Analyzing the correlation between EELV with TTP_Exp_ we found a strong correlation (effect 1.24, CI 95% (1.091; 1.399), *p* < 0.01) (Figure 2) and a negative correlation to ΔP_L_ (effect −0.98, CI 95% (−1.244; −0.724), *p* < 0.01) (Figure 3). The IAP and BMI correlate significantly with P_EsExp_ (effect 0.175, CI (0.112; 0.238), *p* < 0.01 for IAP and effect 0.26, CI 95% (0.161; 0.355), *p* < 0.01 for BMI) (Figure 4 and Figure 5). We found no significant correlation between IAP and E_CW_ or between BMI and E_CW_. The ΔP_L_ was strongly correlation with ΔP (effect 0.97, CI 95% (0.931; 1.01), *p* < 0.01) (Figure 6). 

## 4. Discussion

The main findings of this study are that (1) even patients without ARDS show severe impairment of lung mechanics, (2) this cannot be ruled out by assessing oxygenation and (3) measurements of EELV or ΔP_L_ allow an individualized approach of PEEP management.

We were not targeting the most severe patients. By definition, patients with ARDS fulfilled the criteria for a mild situation. According to the ARDS network table, PEEP would be around 5 cm H_2_O. In the other two groups, P/F ratios were relatively similar to the ARDS group. Although P/F ratio in our three groups did not differ, we found significant differences between the groups in absolute EELV as well as EELV/kg pbw and ΔP and ΔP_L_. One clinical feature of ARDS is atelectrauma, which could not be identified by arterial oxygenation in an ARDS experimental model of ARDS [18]. Intratidal recruitment and derecruitment of alveolar structures lead to shear stress [19]. This alveolar instability contributes to early ventilator-induced lung injury and can be attenuated by appropriate PEEP [20]. By increasing the PEEP up to 15 cm H_2_O, ΔP and ΔP_L_ were reduced, and impaired EELV could be optimized in all groups, suggesting potential for recruitment effects. This lung recruitment can reduce mechanical stress resulting from inhomogeneities in the lung, acting as stress raisers [21].

The optimal approach to titrate PEEP is still under debate in patients with ARDS [22] and even more unclear in patients without ARDS. A variety of bedside tools to titrate PEEP exist such as oxygenation response to PEEP [23], change in driving pressure (ΔP) [24], esophageal pressure [9] or lung recruitment/hyperinflation assessed with the electrical impedance tomography [25]. Such individualized ventilation protocols can improve the outcome in patients with ARDS [26] and in severely obese patients [27], as compared to the standard of care.

Global lung mechanics, e.g., airway pressure, provides a poor surrogate of alveolar dynamics during mechanical ventilation [28]. Apart from that, inspiratory pressures are inaccurate surrogates of true transalveolar stretch, as they do not take into consideration the contribution of the chest wall to respiratory system compliance [1]. Available data suggest that decreased driving pressure in ARDS patients improves the outcome of lung-protective ventilation [24,29]. Interestingly, a study has shown that ΔP has no impact on mortality in obese ARDS patients [30]. However, there are limited data on driving pressure in patients without ARDS. Recently, a secondary analysis of the Lung-Protective Ventilation Initiated in the Emergency Department (LOV-ED) trial found that in patients without ARDS, driving pressure is a risk factor for mortality and the development of ARDS [31]. In this study, esophageal pressures were not recorded, so transpulmonary pressure could not be calculated. Interestingly, these patients had a T_V_ of 8.0 mL/kg pbw and a PEEP of only 5 cm H_2_O. This is quite different from our patients without ARDS. We used a much higher PEEP and aimed for a V_T_ of 6 mL/kg pbw. Although in all groups P_Insp_ increased with higher PEEP, in groups A and B even up to 28 cm H_2_O, the resulting ΔP was lower with the higher PEEP.

Reduction of EELV is a typical feature of ARDS [32]. This phenomenon is attributed to alveolar collapse. We found similar impairment of EELV in our group of moderately obese patients. A physiological study with obese patients (BMI 48 km/m^2^) found that negative TPP_Exp_ is responsible for alveolar collapse [33]. In severely obese patients (BMI almost 60 kg/m^2^), titration of PEEP according to the PEEP/F_i_O_2_ table resulted in a low P/F ratio, elevated driving pressure and impaired lung compliance [34]. On the other hand, applied PEEP may also contribute to VILI by over-distending aerated lung areas. In these critically ill obese patients with ARDS, titration of PEEP according to the best lung function decreased lung overdistension and alveolar collapse assessed by the electrical lung tomography [34]. The optimal PEEP was found with TPP_Exp_, which was slightly positive at 1 cm H_2_O. In our study, BMI was much lower but showed exactly the same physiological alterations with reduced EELV and high ΔP. In our obese patients TPP_Exp_ values were still negative even with a PEEP of 15 cm H_2_O. Although with a PEEP of 15 cm H_2_O, we obtained significantly better EELV and EELV/kg pbw, we could not rule out that much higher PEEP levels might be even more protective. Recent recommendations emphasize accepting plateau pressure higher than 30 cm H_2_O in this context [35]. Interestingly, even in the non-ARDS and non-Obese group, TPP_Exp_ was negative at a PEEP of 5 cm H_2_O in our study.

We used a linear mixed model to evaluate the influence of lung mechanics, intraabdominal pressure and body mass index. The analysis showed a strong positive correlation between EELV and TTP_Exp_ and a negative correlation with ΔP_L._ This means that higher EELV leads to lower ΔP_L_, one cause of VILI, in our study. The ΔP_L_ had a strong correlation to ΔP, which may mean that driving pressure is a parameter of lung mechanics, which is just as accurate, yet more measurable as ΔP_L_, which only can be measured with an esophageal catheter.

In all three groups, E_RS_ and E_L_ decreased with increasing PEEP, but E_CW_ remained the same in all groups. This was most probably due to the best compliance and the lowest E_L_ with the PEEP increasing up to 15 cm H_2_O. Furthermore, we found a positive correlation between BMI and IAP with P_EsEndexp_ in our study. No correlation could be found between BMI and IAP with E_CW_. These findings are in line with the studies in patients with ARDS. Krebs et al. applied different PEEP levels (up to 20 cmH_2_O) in 20 patients with ARDS; one half of this group had elevated IAP, and the other half did not (with a mean IAP of 16 and 8 mmHg, respectively) [36]. PEEP was found to decrease E_RS_ by decreasing E_L_ without influencing E_CW_ in both groups. These findings could explain the fact that patients with higher BMI and elevated IAP are at risk for developing atelectasis, which is one of the major components for VILI, so they may profit from an individual PEEP management approach.

## 5. Limitations

Some limitations in our study must be addressed. First, the study population is relatively small, and we did not analyze outcomes in these cohorts. The physicians at the bedside were not blinded, so that might lead to relative high PEEP even in the non-ARDS groups. We measured P/F ratio on the designated PEEP level and not on a standardized PEEP level of 5 cm H_2_O. We analyzed only PEEP from 5 to 15 cm H_2_O and therefore cannot rule out that even higher PEEP may be optimal.

## 6. Conclusions

Lung mechanics are significantly impaired in ARDS and non-ARDS patients. Combinations of PEEP and F_I_O_2_ to maintain targeted saturation (88–95%) or P_a_O_2_ (55–80 mm Hg), which are used in the ARDS Network studies, do not address individual lung mechanics. With an individualized approach, measurements of TPP and EELV seem to be reasonable and allow a better lung protective approach even in non-ARDS patients.

## Figures and Tables

**Figure 1 jcm-09-02446-f001:**
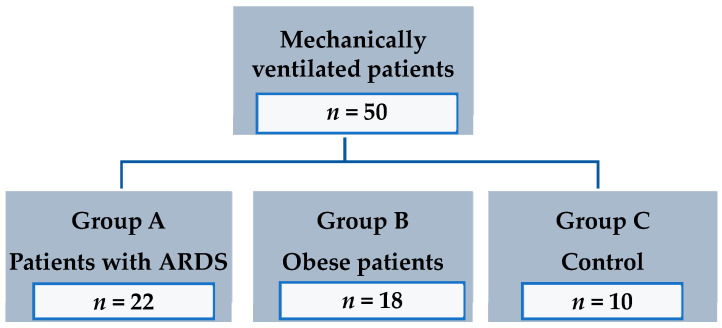
Flowchart displaying patients included in the study.

**Figure 2 jcm-09-02446-f002:**
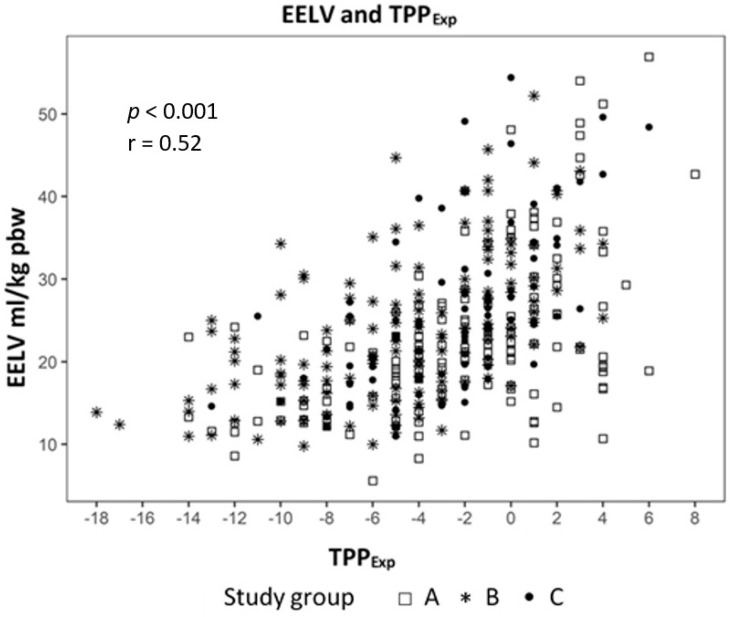
Correlation between end-expiratory lung volume (EELV) and transpulmonary expiratory pressure (TPP_Exp_).

**Figure 3 jcm-09-02446-f003:**
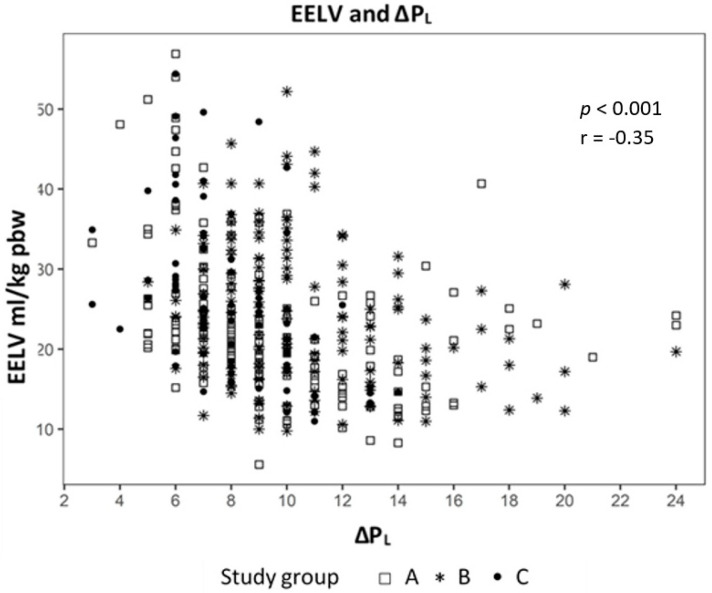
Correlation between end-expiratory lung volume (EELV) and transpulmonary pressure (ΔP_L_).

**Figure 4 jcm-09-02446-f004:**
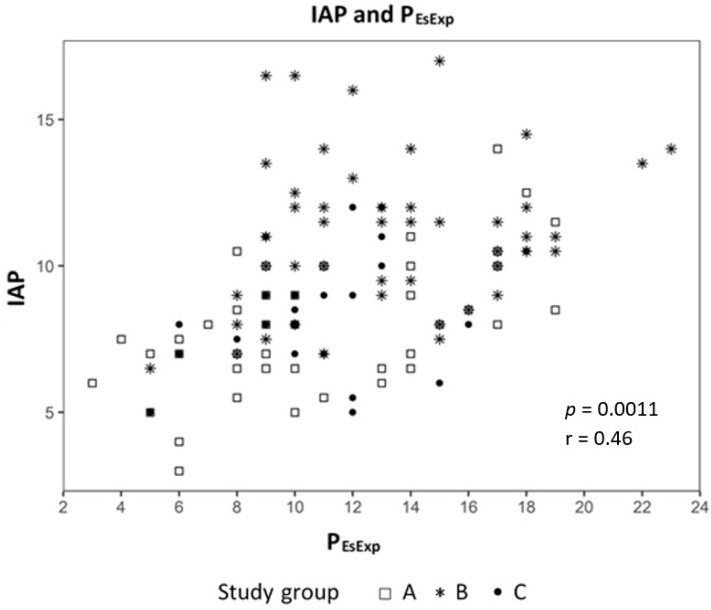
Correlation between intra-abdominal pressure (IAP) and transpulmonary expiratory pressure (P_EsExp_).

**Figure 5 jcm-09-02446-f005:**
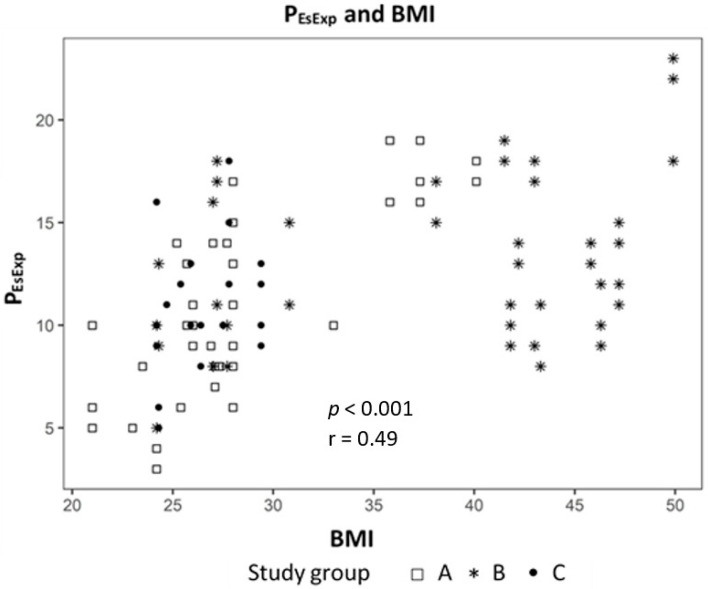
Correlation between body mass index (BMI) and transpulmonary expiratory pressure (P_EsExp_).

**Figure 6 jcm-09-02446-f006:**
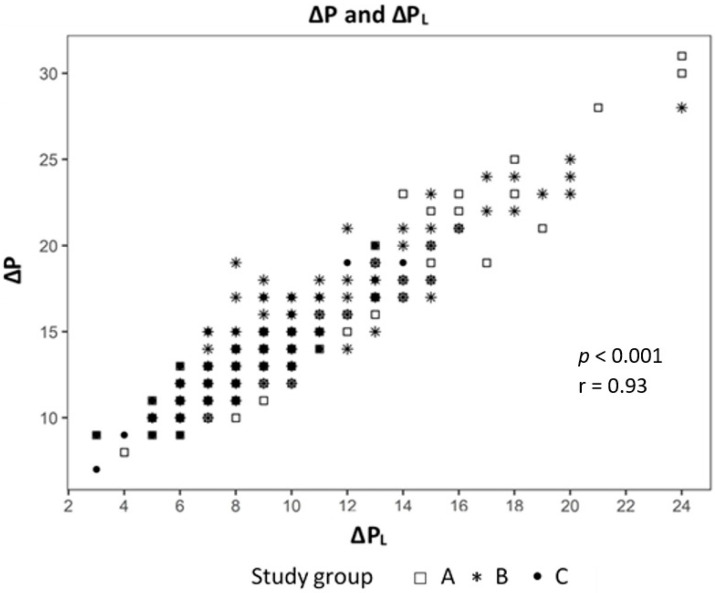
Correlation between driving pressure (ΔP) and transpulmonary pressure (ΔP_L_).

**Table 1 jcm-09-02446-t001:** Baseline characteristics of the study cohort.

	Group A (*n* = 22) Patients with ARDS	Group B (*n* = 18) Obese Patients	Group C (*n* = 10) Control
Demographics			
Age	64 ± 3	72 ± 3	72 ± 3
Height (cm)	173 ± 2	169 ± 3	175 ± 2
Weight (kg)	87 ± 4 *	107 ± 6 ^+^	81 ± 2
BMI (kg/m^2^)	29 ± 1 *	38 ± 2 ^+^	27 ± 1
Male	68%	50%	90%
Cardiovascular presentation			
MAD (mmHg)	78 ± 2	76 ± 2 ^+^	84 ± 3
HR (bpm)	85 ± 4	85 ± 3	87 ± 4
S_p_O_2_ (%)	96 ± 1^#^	98 ± 0	98 ± 0
etCO_2_ (mmHg)	42 ± 1^#^	41 ± 1	37 ± 2
Norepinephrine μg/kgKG/min	0.24 ± 0.05	0.17 ± 0.03	0.24 ± 0.06
Ventilatory parameter			
F_i_O_2_ (%)	44 ± 2 *^,#^	39 ± 1	38 ± 1
V_T_ (mL)	424 ± 6	414 ± 6	433 ± 6
V_T_ (mL/kg pbw)	6 ± 0 *	7 ± 0 ^+^	6 ± 0
RR (/min)	19 ± 0 ^#^	19 ± 0 ^+^	18 ± 0
MV (L/min)	9 ± 0	9 ± 0	8 ± 0
PEEP (cmH_2_O)	14 ± 1 ^#^	15 ± 1^+^	11 ± 1
P_Insp_ (cmH_2_O)	27 ± 1 *^,#^	30 ± 1^+^	24 ± 1
∆P (cmH_2_O)	12 ± 1 *	15 ± 1	13 ± 0
C_Stat_ (cmH_2_O) at PEEP 5 (cmH_2_O)	60 ± 20	58 ± 22	62 ± 23
Gas exchange			
pH	7,3 ± 0 ^#^	7,3 ± 0^+^	7,4 ± 0
p_a_CO_2_ (mmHg)	59 ± 2 *^,#^	51 ± 2	50 ± 2
p_a_O_2_ (mmHg)	93 ± 3 *	106 ± 3	96 ± 5
P/F ratio	210 ± 18	244 ± 15	232 ± 11

BMI = body mass index; MAD = mean arterial pressure; mmHg = millimeter of mercury; HR = Heart rate; bpm = beats per minute; S_p_O_2_ = saturation of oxygen; etCO_2_ = end-expiratory carbon dioxide; F_i_O_2_ = inspired oxygen concentration; V_T_ = tidal volume; pbw = predicted body weight; RR = respiratory rate; MV = minute ventilation; PEEP = positive end-expiratory pressure; P_insp_ = peak inspiratory pressure; ∆P = driving pressure; C_Stat_ = static lung compliance; p_a_CO_2_ = partial arterial pressure of carbon dioxide; p_a_O_2_ = partial arterial pressure of oxygen; P/F ratio = ratio between partial arterial pressure of oxygen to F_i_O_2_. * *p* < 0.05 group A vs. group B, ^#^
*p* < 0.05 group A vs. group C, ^+^
*p* < 0.05 group B vs. group C.

**Table 2 jcm-09-02446-t002:** Respiratory mechanics with positive end-expiratory pressure (PEEP) 5, 10 and 15 cmH_2_O.

	PEEP	Group A (*n* = 22) Patients with ARDS	Group B (*n* = 18) Obese Patients	Group C (*n* = 10) Control
Respiratory mechanics				
P_Insp_ (cm H_2_O)	5	20 ± 1 *	23 ± 1 ^+^	19 ± 1
	10	23 ± 0 *	25 ± 1 ^+^	22 ± 0
	15	28 ± 0 *^,§^	30 ± 0 ^+,§^	27 ± 1 ^§^
∆P (cm H_2_O)	5	15 ± 1 *	18 ± 1 ^+^	14 ± 1
	10	13 ± 0*	15 ± 1 ^+^	12 ± 0
	15	13 ± 0 *^,§^	15 ± 0 ^+,§^	12 ± 0 ^§^
∆ P_L_ (cm H_2_O)	5	10 ± 1 *^,#^	13 ± 1 ^+^	9 ± 0
	10	9 ± 0 *^,#^	10 ± 0 ^+^	7 ± 0
	15	9 ± 0 *^,#,§^	10 ± 0 ^+,§^	7 ± 0^§^
TPP_Exp_ (cm H_2_O)	5	−5 ± 1 *	−8 ± 1 ^+^	−6 ± 1
	10	−2 ± 1 *	−4 ± 0 ^+^	−3 ± 1
	15	1 ± 0 *^,§^	−1 ± 0 ^+,§^	1 ± 0 ^§^
TPP_Insp_ (cm H_2_O)	5	5 ± 0 ^#^	5 ± 1	4 ± 0
	10	7 ± 0 ^#^	6 ± 0	5 ± 0
	15	10 ± 1 ^#,§^	9 ± 0 ^§^	8 ± 1 ^§^
P_EsExp_ (cm H_2_O)	5	10 ± 1 *	13 ± 1 ^+^	11 ± 1
	10	12 ± 1 *	14 ± 1 ^+^	13 ± 1
	15	14 ± 0 *^,§^	16 ± 0 ^+,§^	14 ± 0 ^§^
EELV (ml)	5	1242 ± 85	1181 ± 64 ^+^	1440 ± 101
	10	1541 ± 108 ^#^	1452 ± 73 ^+^	1880 ± 138
	15	1850 ± 119 ^#,§^	1915 ± 97 ^+,§^	2214 ± 154 ^§^
EELV (ml/kg pbw)	5	18 ± 1 ^#^	19 ± 1	21 ± 1
	10	22 ± 1 ^#^	23 ± 1	27 ± 2
	15	27 ± 1 *^,§^	31 ± 1 ^§^	30 ± 2 ^§^
E_RS_ (cm H_2_O/mL)	5	32 ± 3 *	42 ± 5 ^+^	36 ± 6
	10	30 ± 1 *	38 ± 2 ^+^	28 ± 1
	15	31 ± 1 *^,§^	36 ± 1 ^+,§^	29 ± 1 ^§^
E_L_ (cm H_2_O/mL)	5	24 ± 3 *	32 ± 5 ^+^	22 ± 3
	10	21 ± 1 *^,#^	26 ± 1 ^+^	17 ± 1
	15	21 ± 1 *^,#,§^	24 ± 1 ^+,§^	17 ± 1 ^§^
E_CW_ (cm H_2_O/mL)	5	10 ± 1 *	12 ± 1	11 ± 1
	10	10 ± 0 *	12 ± 0	11 ± 0
	15	11 ± 1	12 ± 1	12 ± 1
IAP (cm H_2_0)	5	8 ± 0 *	11 ± 0 ^+^	9 ± 0
	10	8 ± 0 *	11 ± 0 ^+^	9 ± 0
	15	8 ± 0 *	11 ± 0 ^+^	9 ± 0

P_Insp_ = peak inspiratory pressure; ΔP = driving pressure; ΔP_L_ = transpulmonary pressure gradient; TPP_Exp_ = expiratory transpulmonary pressure; TPP_Insp_ = inspiratory transpulmonary pressure; P_EsExp_ = end-expiratory esophageal pressure; EELV = end-expiratory lung volume; EELV/kg = end-expiratory lung volume per kg bodyweight; E_RS_ = elastance of the respiratory system; E_L_ = lung elastance; E_CW_ = elastance of the chest wall; IAP = intraabdominal pressure; * *p* < 0.05 group A vs. group B, ^#^
*p* < 0.05 group A vs. group C, ^+^
*p* < 0.05 group B vs. group C. ^§^
*p* < 0.05 PEEP 5 vs. PEEP 15.

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
