# Peer review of "Prospective Observational Study to Evaluate the Effect of Different Levels of Positive End-Expiratory Pressure on Lung Mechanics in Patients with and without Acute Respiratory Distress Syndrome"

_jcm, 2020, doi:10.3390/jcm9082446_

Round 1

Reviewer 1 Report

Very interesting study on the effect of PEEP values in critically ill with and without ARDS.

Minor comments:

Please describe in more detaile the linear mixed model used.

Abstract: There are scare data . Please correct to "scarce". Spell-check the entire document.

Author Response

Dear Reviewer 1

thank you very much for your comments.

Your minor comments were:

  1. Please describe in more details the linear mixed model. You can find a more detailed description now in Paragraph 2.4 Statistical Analysis. page 5 lane 154
  2. We spell-check once again the final manuscript.

Reviewer 2 Report

There is ongoing scientific enquiry into the best way to set PEEP to reduce both atelectotrauma and at the other end baro/volutrauma which leads to stress and strain. The end result being a transduction of mechanotrauma to biotrauma as well as augmenting pre-existing biotrauma.

Various multicentre trials recently have failed to translate mechanistic studies to clinical benefit including trials using oesophageal pressure as well as lung protective ventilation (without personalised approach) to reduce VILI.

In this paper, Fiedler et al attempt to answer some of the questions both in ARDS and obese patients.

There are some pearls of knowledge to be gained in their experiments including ΔPL and ΔP shown to co-relate well and TPPExp being negative at a PEEP 5 in cohort C. The million dollar question is if this leads to worse outcome?

I have a few issues with the methodology and presentation of results

  1. There are loads of values and acronyms. I would suggest removing the non-significant results to an online supplement as it is difficult to read and interpret the results with some many variables.
  2. The images can be significantly improved 
  3. SPSS has been used for analysis and GraphPad is use for the figures. I presume that the stats are not run again in Graphpad as I thought Graphpad generates figures based on the stats run on its program.
  4. The 3 groups are referred using various terms - use one consistently
  5. Page 3 sentence 78 has called the 3 groups as a priori sub groups. These are not sub groups but just 3 cohorts.
  6. The rationale for using student t test when comparing means of 3 or more groups is not recommended - suggest statistical review to see whether ANOVA or non-parametric equivalent should be used.
  7. Explain the linear mixed methods in more detail
  8. I am unsure of the number of days of testing. Is this for 5 days? This does not come out clearly till page 5. How many drop outs as the days progress as with mortality and extubation, the investigators might end up with a pre-selected group of patients with similar resp mechanics rather than 3 disparate groups.
  9. 399 values - what does this mean. 399 separate instances of testing or 399 individual resp parameters? What is the significance of mentioning this?
  10. This is a mechanistic study of 3 different PEEP levels in 3 groups of patients based on likely differences in resp mechanics. This is not an interventional study assessing outcomes. So why was a longitudinal observation over a series of days conducted rather than a single timepoint within a set period after intubation for eg: within 24 hrs in patients in their first 24 hrs of intubation, undertaken? Why the criteria of 48hrs of intubation for the control cohort. The dynamic changes over 5 days will be missed as well as the issues of point 6 above also arise. 
  11. The use of cardiac arrest patients as control group raises some concerns. Neurological is probably fine as long as they do not have neurogenic pulmonary oedema. There is also no difference between the baseline p/f ratio between control and ARDS.  The obese also have normal lungs and their p/f ratio should also be closer to control. This is likely because, the ARDS cohort is too mild and if so the validity of them as a separate cohort is brought into question. Do we have baseline compliance or Murray lung injury score at baseline? Any reason why the baseline PEEP in the controls is 11 cm H2O. This is not consistent with normal practice.
  12. Need to provide data on how many measurements were cancelled as it might be significant. Probably the patients with the worst impairments failed the interventions and hence couldn't provide the data.
  13. Add a statement of how this could be taken forward if the above issues are addressed

Author Response

 Reviewer

thank you very much for your extensive feedback on our manuscript.

Your comments were:

  1. There are loads of values. You suggest to remove the non-significant results to an online supplement to allow easier reading and interpretation the relevant data. Indeed we have many results from the 50 patients. We tried to describe in the text only the, from our side, most important values. On the other side we include all significant different values in the two tables (table 1 and 2) to give the reader the option to see all relevant data in the main manuscript without a need to move to an online supplement. Please also have in mind that all data in table 2 showing significant differences. So we would like to leave data presentation without a modification.
  2. The images can be improved. We try to improve the images. You will find the new images at page 9 and 10.  Please also see the deleted graphics at the end of the manuscript. These images are not mentioned in the text. Their upload was a mistake during the first version of the manuscript.
  3.  Which program was used for statistics and which for the images. We used SPSS for statistics. We run statistics again in Graphpad to generate the images. On page 5 lane 151 we described "Statistical graphs were developed using GraphPad". We would like to leave it like this.
  4. The 3 Groups are referred using various terms - use one consistently. Thank you we rechecked this point and have now consistent terms on various position in the paper.
  5. Page 3 sentence 78 has called "Sub Groups". As suggested we change that to the term cohorts.
  6.  You recommend to use ANOVA for statistic analysis and to see a statistical review. Indeed, we did ANOVA Tests for the 3 groups showing significances (data are not included in the manuscript). As fare as this approach only shows that there is a difference between the three groups but not between which group we used t-test to find out differences between two groups. This is from a clinical side more interesting we thought. Please have in mind that one of the authors of the manuscript is an expert in biomedical statistics and informatics.
  7. A more detailed description of the mixed linear model was recommended. We include a more detailed description now on page 5 starting from lane 142.   
  8. The days of testing seems to be unclear. Thank you for this advice. Indeed we had some unclear description of the study here and it is confusing. See also comment 9 and 10 for this issue. We tested up to day 3 in this study. We now mentioned this study time earlier in the manuscript on page 3 lane 70 as recommended by the reviewer.
  9. You asked what 399 values means. One value is the the measurement of all relevant lung mechanic parameters to the designated PEEP value. So a Patient on day 1 should get 3 values. One at PEEP 5, 1 at PEEP 10 and 1 at PEEP15. That means for 50 patients 3 values per day for 3 days should be 450 values. Or in other words we are missing 51 values. We include this results now on page 7 lane 183. We also include a table for the data sets as supplemental table.
  10. Indeed, we were not assessing outcome in our study. We include this now as a limitation at the end of the discussion on page 13 lane 307 10 a) The question was why we did a longitudinal observation rather than a single timepoint e.g. x hours after intubation. We recently published in ARDS Patient that no relevant changes in EELV could be found until day 7 after study inclusion (Kalenka et al; Lung 2016 194(4): 527-34.). As mentioned in Comment 9 we measured the patients until day 5 of MV in this study. Dynamic changes might also not always leads to "better" values, it might get also in the other direction. 10 b) Secondly you ask why we had the criteria for 48 hours of intubation.   The criteria of 48 hours of intubation were described misleadingly. We included patients with suspected duration of MV longer than 48 Hours. Also inclusion was latest 24 h after Intubation. These sentences are now changed in the manuscript on page 9 lane 3 lane 77.
  11. Yes as mentioned in the dicussion we had not the severe ARDS patients and neither the severe obese patients. We think that excactly that is one of the most interestingly point in our study. Nonetheless the patients fulfilled the criteria of ARDS as mentioned in the text and shown in table 1. We included here the lung compliance at baseline as suggested (see Table 1). We think that including the Murray Score at baseline might not be helpful as it is not a standard score in Non-ARDS patients. We are also aware the fact that a PEEP of 11 in control patients is probably uncommon. We mentioned that fact in our limitations paragraph already (see page 13 lane 307). This is probably coming from our not-blinded study protocol. Our data in that group suggest that exactly this approach might be useful when a personalized approach is used.
  12.  You asked how many values are missing. Already answered in comment 8 to 10. We are missing 51 out of 450 values. We include this results now on page 7 lane 183. We also include a table for the data sets as supplemental table
  13. We highly appreciate the comments from the reviewer. We think that these significantly improved our manuscript. 

Reviewer 3 Report

The authors report the different volume of ventilation and the lung function in ARDS patients with or without obesity. It is a nice clinical study.

Concerns:

  1. The title is too long could be in a concise writing.
  2. As mentioned by the authors, the population studied is small in each group.
  3. The correlation value could be included in figure legends.

Author Response

Dear Reviewer 3

thank you very much for your work and the feedback on our paper.

Your comments were:

  1. The title is too long. Find attached the new title with is now shorten on page 1 first lanes.
  2. You mentioned that the study population is small. Indeed we only include 50 patients and mentioned that in our limitations paragraph.
  3. You suggested to include correlation values in figure legends. Please find attached new figures including the mentioned values. We calculate these R values with R software and describe that in the M&M paragraph (page 5 lane 164). We also include p values in the graphics.

Round 2

Reviewer 2 Report

The authors have addressed the various comments.

I would suggest adding a comment that there is no change in the results when using ANOVA. I would also suggest stating that the statistics package used is Graphpad PRISM or that it was re run using PRISM and there was no difference in the results.

One comment which is not addressed is the control group including cardiac arrest patients - Did they have normal ECHO post arrest, was there any evidence of elevated PA pressure/CXR findings that might impact respiratory mechanics.

There is no option to say English language is fine in the reviewers' comments and so I have said minor check required.

Many thanks

Author Response

Dear Reviewer,

thanks a lot for your re-review of our manuscript.

We include that we used ANOVA and in case of significance a t-test to analyze differences between two paired Groups. Indeed that is most probably the correct way to describe the process (page 5 lane 148).

We describe the process of developing the graphics accordingly as suggested by you (page 5 lane 163).

Thanks a lot for the advice of exclusion of patients with suspected elevated PA pressure. This was exactly the way we did it. We include this now more specified in the exclusion criteria part (page 3 lane 82).

Many thanks

Armin Kalenka